# Efficacy of EDTA and HEDP Chelators in the Removal of Mature Biofilm of *Enterococcus faecalis* by PUI and XPF File Activation

**DOI:** 10.3390/dj9040041

**Published:** 2021-04-09

**Authors:** Alejandro Álvarez-Sagües, Nerea Herce, Ulises Amador, Francisco Llinares-Pinel, Estanislao Nistal-Villan, Jesús Presa, Laura Álvarez, Magdalena Azabal

**Affiliations:** 1Dentistry Department, Faculty of Medicine, San Pablo CEU University, 28668 Madrid, Spain; nerehrs@gmail.com (N.H.); magdalena.azabalarroyo@ceu.es (M.A.); 2Departamento de Química y Bioquímica, Facultad de Farmacia, Universidad San Pablo-CEU, Urbanización Montepríncipe, Boadilla del Monte, 28668 Madrid, Spain; uamador@ceu.es; 3Microbiology Section, Departamento CC, Farmacéuticas y de la Salud, Facultad de Farmacia, Universidad CEU San Pablo, Boadilla del Monte, 28668 Madrid, Spain; fllipin@ceu.es (F.L.-P.); estanislao.nistalvillan@ceu.es (E.N.-V.); 4Independent Researcher, 28668 Madrid, Spain; jesus_l_presa@yahoo.es

**Keywords:** *Enterococcus faecalis*, EDTA, ethydronic acid (HEDP), passive ultrasonic activation (PUI), Endo-XP finisher (XPF)

## Abstract

Background: Biofilm removal from the root canal during endodontic treatment is necessary to prevent further complications. Irrigation is essential to success. Several irrigants have been proposed without a proper comparison. The aim of the study is to compare the antibacterial capacity of different activated irrigants using passive ultrasonic activation (PUI) or XP-Endo finisher (XPF). Methods: A total of 100 instrumented teeth were incubated in an Eppendorf tube containing 0.5 McFarland of *Enterococcus faecalis* and incubated for 2 weeks at 37 °C. Roots were divided into 5 groups (*n* = 20) according to the irrigant type: ethylenediaminetetraacetic acid (EDTA) (17%), ethydronic acid (HEDP) (9%) mixed with 5.25% sodium hypochlorite (NaOCl), EDTA (17%) mixed with 5.25% NaOCl, PBS, and a control group. Each group was divided into two subgroups (*n* = 10): PUI and XPF. Results: As compared to the untreated control group, the irrigators included in the study had a significant effect in bacteria reduction. The obtained results show HEDP to be the most effective irrigant, since no bacteria were recovered after treatment of this group, followed by EDTA mixed with NaOCl and, finally, the EDTA-irrigated group. Conclusions: HEDP is the best irrigating agent in combination with XPF or PUI file activation to eliminate bacteria in our experimental model.

## 1. Introduction

Most endodontic infections are caused by bacteria [1]. The bacteria that colonize root canals usually form complex communities considered as biofilm. These biofilm structures, contain an extracellular matrix of secreted polysaccharides or extracellular polymeric substance (EPS), hosting different bacteria species with a preference for necrotic tissues [2].

One of the main causes of endodontic failure is the presence, multiplication, and migration into the periapical tissues of biofilms present in the root canals [2]. *Enterococcus faecalis* is a Gram-positive facultative anaerobic bacterium often found in endodontic failures [3]. Cleaning and disinfection of the root canal system is essential to prevent reinfection. This is achieved by bacteria removal using mechanical instrumentation of the root canal combined with specific irrigants.

Instrumentation allows the main canal to be cleaned and shaped, but the root canal system is so complex that not all areas can be accessed. This is due to the presence of isthmic branches, deltas, and accessory canals. In addition, mechanical instrumentation generates a layer of organic and inorganic debris on root canal walls known as smear layer, which hinders their cleaning. Sodium hypochlorite (NaOCl) is the most commonly used irrigant. It acts by both dissolving organic tissues and destroying microorganisms [4,5]. NaOCl can be used at different concentrations, times, volumes, and temperatures [6].

However, NaOCl is not completely effective in dissolving inorganic material or removing the smear layer in areas with difficult access and therefore needs assistance by combination with other irrigators. Chelating agents—normally acidic substances—whose action mechanism is the capture of metal ions, act directly to dissolve the smear layer attached to the canal walls. They can also capture calcium ions from dentin, softening the dentinal walls and thus facilitating the cleaning and instrumentation of the root canal.

Among the chelating agents, EDTA is most used in endodontic treatment, being employed to lubricate root canals, especially calcified teeth, and in the final irrigation protocol alternating with NaOCl. Chelating agents promote the detachment of biofilms from the dentinal walls [7], which favors the reduction of microorganisms and the capture of metallic ions hindering bacterial nutrition [8,9].

In 2005 a concept called continuous chelation was introduced. In continuous chelation, the association of NaOCl with a chelator is used continuously throughout the biomechanics of root canals [7,8,10]. The chelators used in these cases were EDTA and ethydronic acid (HEDP) [7,10,11]. HEDP is an alternative to EDTA as it acts at a more basic pH (around 11) favoring, when combined with NaOCl, a greater concentration of free chlorine, dissolving organic material and increasing the antimicrobial effect of the medium [11].

Both irrigants, NaOCl and the chelating agent, must be activated by ultrasonic activation (PUI) to be more efficient. This improves the dispersion of root canal irrigants via cavitation bubble implosions and acoustic streaming [6].

The XP-Endo finisher file (XPF) [12] has been created to be used in the final irrigation protocol. This file operates at a lower vibration (0.16 Hz) as compared to PUI (30 Hz) [13]. Vibrating XPF files are flexible to move within the root canals using a cyclic winding movement. They can remove the smear layer and promote the entry of the irrigant into the dentinal tubules and, therefore, it can be considered as an alternative to PUI in the activation of the irrigant [12,13].

In this study, the antimicrobial activity of different commonly used irrigants such as EDTA, NaOCl in combination with EDTA, and NaOCl in combination with HEDP was evaluated using standardized *Enterococcus faecalis* biofilms in dental roots as a model. Activation of the irrigants using PUI or XPF was compared to determine the best biofilm removal strategy.

## 2. Materials and Methods

### 2.1. Study Design

All the material used in the study was autoclaved or manually disinfected to prevent external contamination. Teeth extracted for different causes that satisfied the inclusion criteria were obtained from private dental clinics and saved for the analysis. The inclusion criteria for selecting dental pieces considered the use of permanent teeth without previous root alterations or endodontic treatment, confirmed by means of previous radiographs before their selection. All the material preparation was performed and randomly classified by a single operator. Both uniradicular and multiradicular roots were chosen. Each crown was removed by using a 17 mm diamond disc [6,14]. Each root was varnished with nail polish to avoid external contamination. Afterwards, biomechanical preparation was performed by using 10 and 15 k-file instruments to achieve apical patency. After that, a Wave One Gold Primary rotary file was used to finish the instrumentation [15]. Finally, each sample was individually placed in an Eppendorf tube containing 500 µL of brain and heart infusion (BHI) medium (Oxoid) with the coronal opening facing upwards to facilitate subsequent inoculation. They were then taken into an autoclave and sterilized at 121 °C and 1.2 kg/cm^2^ for 20 min.

### 2.2. Ethical Approval

The protocol for this study was approved by the Ethics Committee of the Universidad San Pablo CEU 341/19/15. This protocol implies the use of patient’s dental material for the in vitro experimental study. The length of the study was two years.

### 2.3. Data Collection

#### 2.3.1. Biofilm Preparation and Inoculation

##### Direct Contact Test in the Plates

*Enterococcus faecalis* CECT 481 was initially cultured in Slanetz–Bartley agar plates containing tetrazolium chloride (Oxoid catalogue number: PO5018A). A fresh pass from a single colony in a Slanetz–Bartley agar plate was made the day before used to incubate in tissue culture plates (Figure 1) or inoculation into the dental material to have fresh bacteria cultures.

The study to determine the direct effect of irrigant treatments on bacteria was performed by using 1 mL of a fresh preparation of *E. faecalis* 0.5 McF diluted 1/10 in Eppendorf tubes. Bacteria were then centrifuged for 5 min at 1000 rpm in a microcentrifuge. Supernatant was removed and bacterial pellet was resuspended and incubated for 1 min in 1 mL of the different irrigants under study. Following incubation, bacteria were centrifuged again for 1 min at 1000 rpm and the supernatant was discarded. The process was repeated twice with PBS to remove remaining irrigant traces. Finally, the bacteria were resuspended in 1 mL of PBS and a 1/10 serial dilution was performed before plating a 100 µl volume of each dilution to determine the original bacteria concentration. Incubation of Slanetz–Bartley plates at 37 °C was carried out for 2 days before counting the number of bacteria colonies.

The preparation of pure free biofilms was achieved in tissue culture plates by incubating 1 mL of *E. faecalis* 0.25 McF in individual wells of a 12-well tissue culture plate. Plates were sealed using parafilm and incubated for 2 weeks at 37 °C before analysis. After two weeks, bacteria biofilms in the tissue culture plates were carefully recovered with a spatula and transferred to individual Eppendorf tubes. To standardize samples, the biofilm from one well was transferred to one Eppendorf tube. Supernatant was removed after a gentle centrifugation (1 min at 500 rpm). Bacteria biofilm was treated for 1 min as indicated in the previous paragraph. After treatment, biofilm was disaggregated by extensive pipetting during the washing steps before dilution and plating to determine bacterial titter as indicated before.

##### Antimicrobial Activity in the Root Canals

The elaboration of the biofilm inside root canals was generated by initially preparing a fresh culture of *E. faecalis* in Slanetz–Bartley agar plates as previously indicated. The following day, fresh colonies were resuspended in phosphate-buffered saline (PBS) until it reached a turbidity of 0.5 McFarland units. Subsequently, 50 µL of this preparation were used to inoculate individual Eppendorf tubes containing 450 µL of BHI and one dental root that was prepared as indicated above. After inoculation, tubes were closed, and all samples were placed in an incubator at 37 °C for two weeks before the analysis.

#### 2.3.2. Group Selection

After two weeks, teeth were distributed randomly into 4 experimental groups (*n* = 20) and 1 control group (*n* = 20).

In all the experimental groups, the specimens were irrigated for 1 min, divided in 3 cycles of 20 s, administering a total of 3 mL (1 mL per cycle) of irrigant with a monojet syringe and lateral exit needle.

The four irrigants used in the study were 1× PBS (Gibco), 17% EDTA disodium (Sigma Cat# E6635), EDTA (17%) mixed with 5.25% of NaOCl in a 1:1 ratio, HEDP Dual Rinse (Medcem), which was prepared following the manufacturer’s instructions (1 capsule dissolved in 10 mL NaOCl 5.25%). The control group received no treatment. Each group was divided into two subgroups according to the activation method, PUI or XPF.

Passive ultrasonic activation was done at a speed of 1 in a Satelec ultrasound machine, using the Irrisafe 20/21 irrigation tip. XP-Endo finisher file was operated by means of a VDW Silver Reciproc endodontic motor at the speed and torque recommended by the manufacturer: 800 rpm and 1 N cm, respectively.

#### 2.3.3. Collection of Bacteria from Biofilm

After treatment of each individual root, dentin was collected from the apical third of the root canal with a SX- Protaper rotating file, introduced through the apical foramen of each sample to a depth of 2.5–3 mm. Dentin obtained from each tooth was weighted. The average amount of dentin extracted per sample was 15 mg.

Dentin obtained from each individual tooth was introduced into test tubes containing 1 mL of PBS and vortexed for 1 min to extract the bacteria into the medium. Sequential 1/10 dilutions were prepared before plating 100 µL in Slanetz–Bartley agar plates to quantify the recovered bacteria. After 48 h of incubation, individual colony forming units (CFU) from each agar plate were counted in order to determine the relative CFU/(mL × mg of dentin) in each sample. Negative control roots not incubated with *E. faecalis* were used to verify absence of external contamination.

#### 2.3.4. Visualization of Biofilm Formation

As described before in the section “Antimicrobial Activity in the Root Canals”, individual inoculated teeth were collected after 2 weeks of incubation. Samples were sectioned longitudinally with a homemade guillotine to have access to the root canal and visualize biofilm formation inside. To be able to visualize the biofilm under our SEM electron microscope, the cut material was covered with the ionic liquid 1-butyl-3-methylimidazolium bis-(trifluoromethyl sulfonyl) imide (BMIM) (Sigma-Aldrich) for 20 min. The sample was then mounted in the microscope device prior to scanning using a FEI XL30 electron microscope. Images were taken using 10 kV and 2000× magnification.

### 2.4. Data Analysis

The statistical analysis of the different experimental groups was carried out by means of a Kruskal–Wallis test. The Wilcoxon test was used to analyze the differences between pairs using R software. Non-parametric tests were used because of the large asymmetries and tails that were found within the results of each group.

## 3. Results

*Enterococcus faecalis* can be used as a model bacterium to study biofilm in root canals. In order to characterize the behavior of the bacterium used in our studies, we initially tested the effect of different irrigants that are commonly used in cleaning root canals (Figure 1). In Figure 1A, a suspension of 0.5 McF *E. faecalis* diluted 1/10 was used to prepare the different treatments that were incubated for 1 min in 1 mL of the different irrigants under study. We can observe that any of the conditions containing NaOCl were enough to eliminate all living bacteria from the inoculum; however, EDTA treatment alone did not cause any effect on *E. faecalis* growth properties.

Biofilm structures could be important resistance structures that protect bacteria from external abrasion. To study the effect of irrigants on *E. faecalis* biofilm, we developed a method to generate biofilm in tissue culture plates by cultivating a preparation of 0.25 McF of *E. faecalis* in 1 mL of liquid BHI media in independent wells of a 12-well tissue culture plate. After inoculation, the plate was sealed to the lid with parafilm to prevent desiccation and establish long term microaerophilic conditions. Bacteria can form continuous biofilms at the bottom of the plate. In our hands, the best time to analyze these biofilm structures was two weeks after inoculation (data not shown), before the mature biofilm starts to detach and degrade.

Resistance of *E. faecalis* mature biofilm was analyzed against the different disinfection irrigants (Figure 1B). After two weeks of incubation, individual biofilms from the tissue culture wells were recovered, trying to minimize fragmentation. Biofilms were placed in individual Eppendorf tubes and allowed to deposit at the bottom. The media at the top was removed before exposure to the different irrigants for 1 min. Disinfecting irrigants were then removed and biofilm was broken by intensive pipetting. Resuspended bacteria were washed 3 times with PBS before the serial dilutions required for titration in Slanetz–Bartley media. Not a single colony could be recovered from the cultures treated with NaOCl alone or in combination with other substances.

Failure of dental endodontic treatment is commonly caused by bacteria proliferation in areas of the root canal where disinfecting irrigants have difficult access. We have standardized a method to generate *E. faecalis* biofilms in the tooth root, allowing a reproducible comparison between experimental conditions.

In order to visualize the formation of the biofilm, dental roots were inoculated with bacteria as indicated in the material and methods section. After 2 weeks of incubation, roots were sectioned, and the root canal analyzed by scanning electron microscopy (SEM) (Figure 2).

Unfortunately, the initial attempts to visualize biofilm samples under the SEM microscope were unsuccessful. Freezing and vacuum were required in our SEM microscope to obtain an image with good resolution. This approach had to be abandoned since biofilm structures were destroyed during the process. We finally were able to obtain biofilm images by covering the sectioned root area with the ionic liquid BMIM to prevent evaporation during SEM visualization. The presented image has the best resolution that we could obtain using the ion liquid approach. As can be observed on the right-hand side of the image (Figure 2), dentinal tubules of the root canal wall can be occupied by bacteria structures. Some individual bacteria can be observed (0.5–1 µm in diameter), as well as larger continuous structures of up to 10 µm. Individual bacteria can form irregular structures that accumulate forming initial clusters that may evolve into continuous patches coincident with biofilm structures (patches of around 10 µm of diameter).

Once the biofilm conditions were stablished inside the roots, the effect of the different proposed treatments in removing and destroying bacteria in the root canals was characterized. As indicated in the methods section, we stablished a protocol to systematically treat each tooth for 1 min and later recover dentin in order to resuspend the bacteria and be able to quantify *E. faecalis* at the different experimental conditions.

As can be appreciated in Figure 3, the use of the XPF or PUI treatment using PBS, without a disinfecting irrigant, has a negligible bactericidal activity in removing *E. faecalis* as compared to the control group, where the root canal was not cleaned with any method (black bar). The activation technique of the irrigants with XPF or PUI does not influence the results [2], since no significant differences were obtained when comparing the use of one or the other file, despite the irrigant of choice. Unfortunately, four samples were lost due to cross-contamination of dental samples.

An additional observation is that the EDTA chelating effect is not completely effective in helping to remove bacteria; however, file action in EDTA-irrigated samples had an average of 1.68 × 10^2^ CFU/mL compared to the average of 9.82 × 10^3^ CFU/mL in the control group, indicating a cooperative effect of EDTA with the XPF or PUI file cleaning action.

Previous studies indicate that a combination of EDTA with NaOCl is an efficient strategy to remove bacteria from the root canal. In our hands, this treatment was also more efficient that PBS or EDTA alone (Figure 3 and Table 1).

The roots treated with a combination of EDTA and NaOCl showed an average of 1.39 × 10^1^ CFU/mL, which indicates a reduction level of almost 3 logarithmic orders of magnitude (see Table 1: 2.911 for XPF and 2.956 for PUI) which corresponds to almost 99.9% of bacteria reduction in relation to the untreated control group. This condition presents a statistically significant colony decrease in comparison to other groups, but incomplete bacterial removal. Comparison of this combination with the one treated with EDTA alone showed a significant decrease, indicating a cooperative effect of NaOCl and EDTA. This partial removal of bacteria is still unsatisfactory from a clinical point of view.

Finally, in our experimental model, the use of HEDP dissolved in NaOCl is the most efficient method to eliminate remaining *E. faecalis*. No growth was observed in any of the conditions where this irrigant was used.

## 4. Discussion

Previous independent studies have intended to determine the bactericidal effect of different irrigants or different treatments to select the best practice choice. Here, we present a systematic comparison of different state-of-the-art options in removing residual bacteria from biofilms formed at the root canal after endodontic interventions.

In our initial in vitro results, we can appreciate that EDTA does not have a significant bactericidal effect. However, EDTA can have an important contribution in helping to detach biofilm from the dental material. Previous studies support this antimicrobial action [16,17]. Goldman et al. [16] determined that EDTA solution significantly reduces bacteria in necrotic root canal. This could be attributed to the disaggregative activity that EDTA has on the smear layer and, therefore, the disinfectant effect by mechanical dragging [16]. Because of this, EDTA is proposed to remove biofilms adhered to the root canal walls. However, other studies have shown that EDTA has no direct antimicrobial activity since it has little or no effect on bacterial shedding [18,19].

Biofilms are protected by extracellular polymer substance (EPS). EPS consists of water, proteins, polysaccharides, extracellular DNA, and other components [4]. It may also contain metals such as calcium (Ca^2+^), which maintain the stability, architecture, and strength of the biofilm [20]. The EDTA chelating nature acts in directly sequestering Ca^2+^ and other cations, causing disruption of the biofilm [10,20,21,22]. This chelating activity explains the results observed in Figure 3, where the use of EDTA as an irrigant presented a significant bacterium decrease with respect to the control group.

However, the presence of colonies in the plates after treatment in different dilutions showed that the cleaning effect of using EDTA as an irrigant is not enough to completely disinfect the root canals. Enhancement of ETDA antimicrobial activity can be achieved by combining disinfecting agents to ETDA such as cetrimide or antimicrobial peptides [23,24].

It is of interest to mention that EDTA can be presented as a disodium or tetrasodium salt, affecting its properties [10]. The EDTA disodium salt has an acidic close near-neutral pH. The acidification of the medium interacts with the sodium hypochlorite, reducing its activity such as the dissolution of organic tissue and antimicrobial activity. However, the EDTA tetrasodium salt has a basic pH close to 11, which makes this tetrasodium salt compatible with NaOCl without altering its properties, making it a better combination with NaOCl [10,11]. In our study we used EDTA disodium salt combined with 5.25% NaOCl since this form is still used as the most common chelating agent [10,11,17].

From our results, we can propose that the 1:1 combination of NaOCl 5.25% and EDTA 17% (disodium form) is more effective than EDTA alone. This suggests that by optimization of the combination of the two irrigants, we could improve bacteria removal of the two solutions, and in our case, the combination of the two substances was prepared just before the administration of the irrigant.

Ethydronic solution (HEDP) in 5.25% NaOCl is a good alternative to EDTA in the process of continuous chelation. This solution involves the combination of sodium hypochlorite NaOCl and a chelator during the chemical–mechanical preparation [10]. The main difference of HEDP with respect to disodium EDTA is that it is a weak alkaline chelator acting in the pH range of 10.8–12.2, while disodium EDTA operates at lower pH. These alkaline conditions of the ethydronic acid favors its use with NaOCl solutions, not affecting the pH of the NaOCl solution [10,11,25].

The reason for the better NaOCl compatibility with HEDP compared to EDTA may be explained since HEDP is a non-nitrogenous chelator; it contains phosphorus instead of nitrogen. In NaOCl, the chlorine essentially carries a positive charge and will attack the electrophilic centers of the nitrogen atoms [25]. Phosphorus is less electronegative than nitrogen, so it is less likely to react with NaOCl [25]. The results observed in Figure 3 showed a maximal effect by the combination of HEDP with 5.25% NaOCl. Bacterial levels were reduced below detection levels.

Biel et al. [11] compared EDTA tetrasodium salt with HEDP. The results showed that the tetrasodium salt of EDTA has some compatibility with NaOCl but only at low concentrations and in the short term. However, for the intended application in the clinical practice, HEDP seems much more suitable than the EDTA counterpart [10,11]. A further study published by Wright et al. [26] evaluated various chelating agents and their interactions with pH and chlorine, trying to determine which agent had better behavior when combined with NaOCl at different times, pH values, and temperatures, and the results indicated that the ethydronic activity was much higher than in EDTA alone. The irrigation and activation techniques chosen used in this study were passive ultrasonic activation (PUI) and XP-Endo finisher (XPF), as previously described [6,12,13].

If we compare the different forms of activation with articles that have been published, we can appreciate that despite of the concentration of irrigant and different incubation times, both XPF and PUI activation present very similar results. PingPing Bao et al. [12] proposed to combine two techniques, PUI and XPF, with irrigation of NaOCl 3% and EDTA. This resulted in 95% with PUI and 99% with XPF efficacy in bacteria reduction after 4 weeks of incubation. Adham Azim et al. [14] used hypochlorite 6% and EDTA and have similar results with XPF 98% [13] after 3 weeks. B. Bhuva et al. [14] used NaOCl 1% after three days of incubation with PUI and obtained a 99% reduction of bacteria. This means that both activation techniques produce similar results, therefore the type of irrigant applied is of more relevance than the selected activation procedure.

The results obtained in our experimental model indicate that the activation with XPF can achieve similar results to the ones achieved with PUI. The XPF is a relatively new instrument and there are very few studies on biofilm reduction compared to PUI [27]. Suitability of XPF or PUI to the clinical situation should be determined in situ. Future studies will determine whether the nature and complexity of the biofilm may determine the nature of the file and whether HEDP in NaOCl is as efficient as presented here in eliminating bacteria from root canals.

## 5. Conclusions

Current irrigations of the root canal during endodontic treatment are performed using NaOCl alone or combined with other irrigants. The use of 5.25% NaOCl alone is an efficient method to remove infectious agents that may be present forming biofilms. However, NaOCl alone is not always efficient, and requires additional agents such as chelating agents (EDTA or HEDP) to remove root associated biofilm. We present an in vitro experimental setup using extracted teeth to stablish root canal biofilms formed by *E. faecalis* bacteria. Our experimental setup allows comparing the efficacy of different disinfecting protocols to clean root canals, including irrigant activation to improve cleaning. In this experimental model, there were no significant cleaning differences when comparing passive ultrasonic (PUI) or activation by Endo XP finisher (XPF) to activate the irrigant and eliminate bacteria. However, when comparing irrigants, HEDP showed the best results and is the most appropriate chelator to combine with sodium hypochlorite in irrigation protocols. This result suggests that HEDP could be considered as the irrigant of choice in endodontic bacteria debridement in combination with NaOCl.

## Figures and Tables

**Figure 1 dentistry-09-00041-f001:**
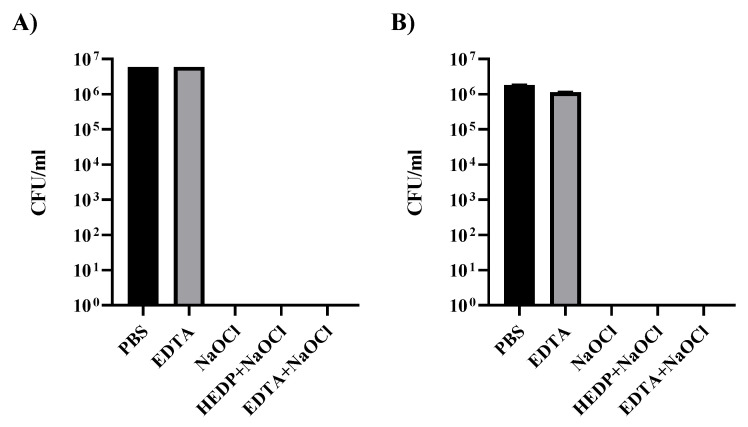
Effect of disinfecting irrigants against *E. faecalis in vitro*. Results of tests with irrigants against bacteria (**A**) and against bacteria forming mature biofilm (**B**). In both cases, no bacteria were detected in treatments using irrigants with NaOCl.

**Figure 2 dentistry-09-00041-f002:**
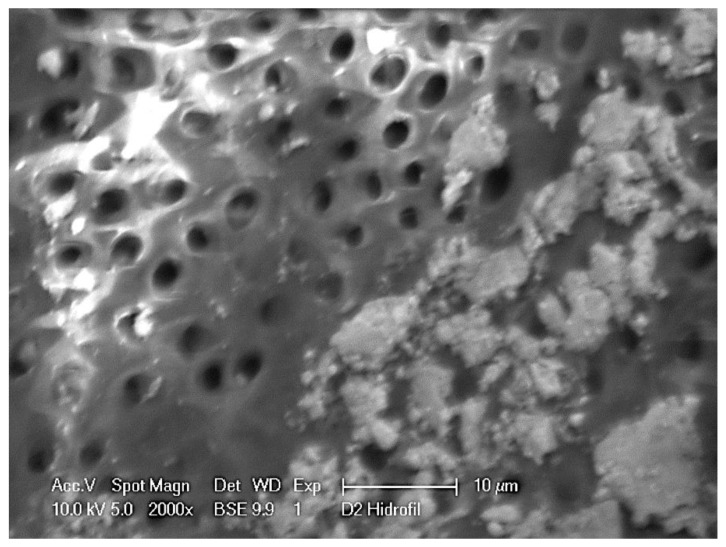
Scanning electron microscopy (SEM) of *E. faecalis* biofilm formation in the root canal. At the left of the image, we can observe the clean dentin wall of the root canal with the access of the dentinal tubules. On the right, individual bacteria and biofilm formation on the surface of the dentin wall. Images were taken 2 weeks after inoculation of the dental material with the bacteria. Access to the area was achieved by a clean cut of the tooth. Image characteristics are indicated within the image.

**Figure 3 dentistry-09-00041-f003:**
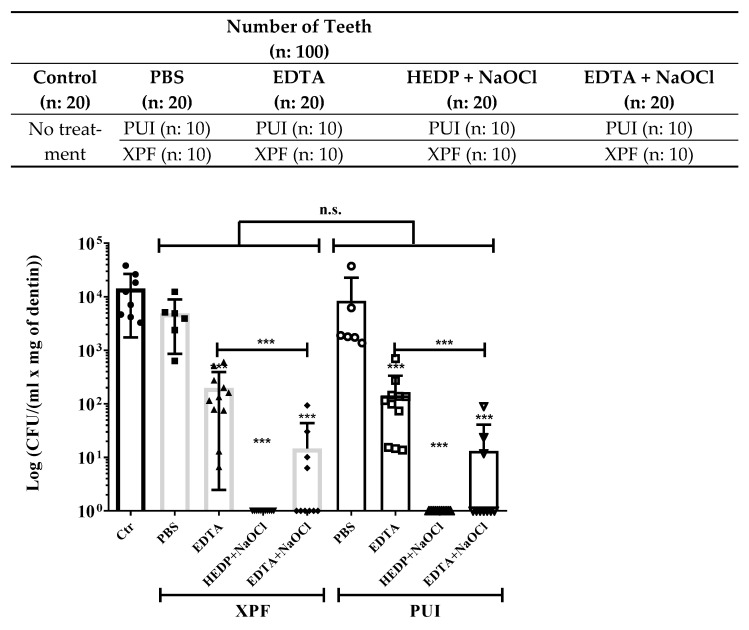
Quantification of *E. faecalis* remaining in root canals after different file and irrigant treatments. **Above**: Classification of the different groups and subgroups used in the study. **Below**: Dental roots incubated for 2 weeks in the presence of *E. faecalis* were treated as indicated for 1 min. Dentin from root canals was recovered and used to quantify remaining bacteria. Control (black bar) untreated group is compared to XP-Endo finisher (XPF) (grey bar) and passive ultrasonic activation (PUI) (white bar) file treatments together with the indicated irrigants *** *p* < 0.001; n.s. (non-significant differences): *p* > 0.01.

**Table 1 dentistry-09-00041-t001:** Mean and exact *p* values obtained from the comparison of the indicated experimental groups with the control (untreated group). Mean and *p* value numbers correspond to Figure 3. Numbers were obtained by Kruskal–Wallis statistical analysis. No differences were appreciated between XPF and PUI. Ethydronic acid (HEDP) irrigation had the highest effect in eliminating bacteria with respect to the control group.

Relative Reduction of Bacteria after Indicated Treatments Compared to Untreated (Control Group)			
	Relative Decrease (in Logarithmic Value) and *p* Value		XPF vs. PUI Comparison
	XPF	PUI	XPF vs. PUI (*p* Value)
PBS	0.386 (5.78 × 10^−2^)	0.152 (4.78 × 10^−2^)	(6.31 × 10^−1^)
EDTA	1.780 (2.49 × 10^−5^)	1.983 (1.56 × 10^−5^)	(4.60 × 10^−1^)
EDTA & 5.25%NaOCl	2.911 (3.62 × 10^−5^)	2.956 (3.33 × 10^−5^)	(7.23 × 10^−1^)
HEDP & 5.25%NaOCl	4.075 (1.05 × 10^−5^)	4.075 (1.05 × 10^−5^)	(1.0)

## Data Availability

Not applicable.

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
