# Peer review of "Efficacy of EDTA and HEDP Chelators in the Removal of Mature Biofilm of Enterococcus faecalis by PUI and XPF File Activation"

_dentistry, 2021, doi:10.3390/dj9040041_

Round 1

Reviewer 1 Report

General comments

The article has improved compared to the first version, however, overall, the manuscript is not thoroughly revised, nor organized well. Each section recommends a fine revision.

Detailed comments

Abstract- Based on the “Dentistry Journal” guideline, “The abstract should be a total of about 200 words maximum. The abstract should be a single paragraph and should follow the style of structured abstracts, but without headings: 1) Background: Place the question addressed in a broad context and highlight the purpose of the study; 2) Methods: Describe briefly the main methods or treatments applied. Include any relevant preregistration numbers, and species and strains of any animals used. 3) Results: Summarize the article's main findings; and 4) Conclusion: Indicate the main conclusions or interpretations”

  • Objective: what’s the PUI? Need to spell it in the beginning
  • Material and method section: indicated there are 5 groups according to the irrigant, but showed only 4?-please clarify. The sentence: “Dentin material recovered from the root canals was used to quantify remaining bacteria.”-not clear and should clarify
  • Results: authors indicated that significant effect which followed by statistical values
  • Conclusion section: writing format, spacing was messed up

Introduction- the sentences and paragraphs are not clearly address the current state of the field of the research, nor address the importance of the proposed research. In fact, it is hard to find the main aim of the proposed research. Not consistently using NaOCl and hypochlorite.   

Materials and methods-

  • Ethics Committee-should indicated subject’s informed consent form received?

  • Preparation of teeth and individualization of samples – describe thoroughly how to collect the teeth. What is the “Biomechanical preparation using the files”?

  • Group selection – clarify the random selection of the teeth? labeling the teeth or blind study? How many operators been involved study? Describe the difference between the PBS and no treatment control. Table 1 should be revised based on the sequence of the first-to fifth group-poorly organized

Results

  • Figure 3- There are no matches between the schematic figure and bar graphs in terms of the experimental conditions. Vertical axis of bar graph should indicate the name of the organism
  • Table 1. p value should be standardized, RINSE represent what?

Discussion and Conclusions: Overall, the paragraphs are not fully developed to support the author’s main idea. Many of sentences are redundant from the introduction which are well-known facts. Could not find any relevant depth of literature review in the discussion section. Based on author’s result, Figure 3, HEDP + NaOCl demonstrated a highest antibacterial effects in the E. Faecalis CFU, however the conclusion reflects that “EDTA disodium salt in combination with NaOCl present a relevant bacterial debridement which is overpassed by the combination of NaOCl with the weak chelating agent HEDP”, which is not accorded.

There are no sections for the Author Contributions, Institutional Review Board Statement, Informed Consent Statement, Data Availability Statement, and Conflicts of Interest. Wished that these were addressed in the separate document.

Reference style should be followed the journal guideline

Author Response

Reviewer 1’s comments:

  1.  General comments The article has improved compared to the first version, however, overall, the manuscript is not thoroughly revised, nor organized well.  Each section recommends a fine revision. Detailed comments Abstract- Based on the “Dentistry Journal” guideline, “The abstract should be a total of about 200 words maximum.

Thank you very much for appreciating the improvement. As requested, we have adjusted the abstract to the requirements.  Lines 17 to 29.

  1. The abstract should be a single paragraph and should follow the style of structured abstracts, but without headings:  1) Background: Place the question addressed in a broad context and highlight the purpose of the study;  2) Methods: Describe briefly the main methods or treatments applied. Include any relevant preregistration numbers, and species and strains of any animals used.  3) Results: Summarize the article's main findings; and  4) Conclusion: Indicate the main conclusions or interpretations”

We appreciate the suggestion, following reviewer´s indications, we have structured the abstract in one patagraph, including the 5 indicated sections. Lines 17 to 29.

  1. Objective: what’s the PUI? Need to spell it in the beginning

PUI acronym means passive ultrasonic irrigation and has been included in line 19. In addition it is better described in lines 68-70.

  1.  Material and method section: indicated there are 5 groups according to the irrigant, but showed only 4?-please clarify.

Thank you for noticing, the text has been corrected, including the four experimental conditions and one control (Lines 143 and 147). In additions the number of groups have been included in the abstract sections (lines 21-24).

  1. The sentence: “Dentin material recovered from the root canals was used to quantify remaining bacteria.”-not clear and should clarify

We apologize for not being clear enough. Description of dentin recovery from the root canals and bacteria quantification is described in lines 153-158.

  1. Results: authors indicated that significant effect which followed by statistical values

We apologize, but we are not sure if the referee is trying to point out why the statistical analysis of data in figure 3 is presented in table 1. The reason is that with the table we wanted to be more precise indicating the exact numbers when performing the statistical analysis. We are sorry if we miss the question and if so, we will try to be more precise in the answer.

  1. Conclusion section: writing format, spacing was messed up Introduction- the sentences and paragraphs are not clearly address the current state of the field of the research, nor address the importance of the proposed research. In fact, it is hard to find the main aim of the proposed research. Not consistently using NaOCl and hypochlorite.

We have changed the conclusion to the reviewer´s suggestion, addressing more clearly the current state of the field, the importance, and the aim of this project. We hope it is clearer now.

  1.  Materials and methods- • Ethics Committee-should indicated subject’s informed consent form received?

Many thanks for your advice. In relation to the informed consent for scientific use provided by the donors as part of the extraction process, in Spain extracted teeth are considered biological material and according to BOE Law 41/2002, of 14 November of 2002, on the autonomy of the patient and the rights and obligations pertaining clinical information and documentation, a donor informed consent is not necessary for the handling of these remains. However, an authorization from the Ethics Committee of the University is always required for handling human samples. In our case, the Ethics Committee protocol CEU 341/19/15 has been approved and it indicated in line 100.

  1. Preparation of teeth and individualization of samples – describe thoroughly how to collect the teeth. What is the “Biomechanical preparation using the files”?

We apologize if we were clear enough. We describe the source of teeth in lines 86-90.  “Private dental clinics that extracted teeth for different causes and that satisfied the inclusion criteria were collected and saved for the analysis.

When we indicate biomechanical preparation, we are referring to the root canal treatment. This treatment has two phases: an initial mechanical one, which is performed with endodontic files aiming to clean the bacteria from the root canal, and the chemical one, that tries to remove remaining bacteria using chemical irrigants to complete the cleaning process.

  1. Group selection – clarify the random selection of the teeth? labeling the teeth or blind study? How many operators been involved study?

We obtained the teeth according to our inclusion criteria. We randomly divided the teeth into groups based on previous studies and their variability.

The teeth were initially included based on the inclusion criteria described in 84-91. Once selected, teeth were pooled and randomly chosen for the assays by a single operator.

  1. Describe the difference between the PBS and no treatment control.

Thank you for noticing, the text has been modified to clarify the groups. In the Control group, there is no treatment at all. In the PBS group, there is treatment, using files, but the irrigant in this case is PBS. They are different groups, we have made the appropriate changes so that there is no confusion between the two groups. Lines 21-24.

  1. Table 1 should be revised based on the sequence of the first-to fifth group-poorly organized

Thank you for the suggestion. We have tried to better explain the table 1 in the Table and table footnote. In this table we are trying to put exact numbers to the data observed in figure 3. Basically, we are comparing the relative bacteria reduction in each of the conditions presented in the table with respect to the Control (untreated group). The relative reduction numbers are presented as logarithmic reduction numbers. The p values are the exact p values comparing the mean of each of the presented conditions with the untreated Control group. P values > 0,1 are represented as non-significant differences. All the data comparison was performed by using Kruskal-Wallis statistical analysis.

  1. Results. Figure 3- There are no matches between the schematic figure and bar graphs in terms of the experimental conditions. Vertical axis of bar graph should indicate the name of the organism.

Sorry for not being precise in the matches. We hope the description is clearer now. Changes in Figure 3 are included in the current version following the reviewer suggestion. Table has been updated with the same strauture of groups as compared to the figure groups.

  1.  Table 1. p value should be standardized, RINSE represent what?

Sorry if we mis the exact meaning of the question. P values represent standardized numbers per se. In this case we are performing a statistical analysis using a Kluskal-Wallis approach. In this approach we can compare different groups under analysis and determine the exact p value as well for each comparison. Those numbers are the ones indicated in Table 1.

We have changed RINSE (the commercial brand name to HEDP, which is the chemical name and included it in the current figure and table versions.

  1. Discussion and Conclusions: Overall, the paragraphs are not fully developed to support the author’s main idea. Many of sentences are redundant from the introduction which are well-known facts. Could not find any relevant depth of literature review in the discussion section.

We apologize that our discussion is not consider fully developed by the reviewer. We have tried to explain some of the issues that we consider are important to understand the results and the message of the manuscript. We will appreciate if the reviewer can be more precise in order to improve some of the details he/she consider that we are missing.

As indicated before, we have change the conclusion section in order to make it clearer. 

  1. Based on author’s result, Figure 3, HEDP + NaOCl demonstrated a highest antibacterial effects in the E. Faecalis CFU, however the conclusion reflects that “EDTA 2 disodium salt in combination with NaOCl present a relevant bacterial debridement which is overpassed by the combination of NaOCl with the weak chelating agent HEDP”, which is not accorded.

Thank you for noticing. Text has been changed accordingly in line 357-360.

  1. There are no sections for the Author Contributions, Institutional Review Board Statement, Informed Consent Statement, Data Availability Statement, and Conflicts of Interest. Wished that these were addressed in the separate document. Reference style should be followed the journal guidelin

Thank you again. Following editorial guidelines, we have introduced the corresponding sections including: author’s contributions, funding, conflicts of interest, institutional Review board statement, informed consent statement, data availability statement and conflicts of interest.

Reviewer 2 Report

The manuscript “Efficacy of EDTA and HEDP chelators in the removal of mature biofilm of Enterococcus faecalis by PUI and XPF file activation” by Alvarez-Sagues et al. compared different methods for removing E. faecalis biofilms from teeth. The different methods tested included 2 types of files to clean the root canal (PUI and XP-Endo) as well as different combinations of irrigants including EDTA, HEDP, and sodium hypochlorite.

Overall, the experiments seem well designed, although there is not enough information provided in the materials and methods section (see comments below). Additionally, several previous studies have found differing results regarding the killing of E. faecalis by NaOCl. The discussion and conclusions would be significantly strengthened if the authors compared and constrasted their results with these previous studies. Why do the authors think they saw such good inhibition of E. faecalis in this study? Was the strain of E. faecalis or the way the assays were set up different than other studies that saw higher rates of resistance?

Minor comment: Multiple examples of spaces missing between words: line 26 (roots amputated), line 14 (address requests for reprints to), line 40 (bacteria in), line 51 (Enterococcus faecalis is a), etc.

Line 22: please define PUI acronym here

Line 36: please define NaOCl abbreviation the first time it is used

Line 49: Many biofilm bacteria don’t “prefer” necrotic tissue

Line 95: Were the teeth removed from patients and prepared for autoclaving immediately? Or were they stored before autoclaving and biofilm assays? Can the authors provide more information on the patients from which teeth were obtained?

Line 113: Was E. faecalis CECT 481 originally stored as a freezer stock? How long werw the bacteria kept on agar plates in total?

Line 116: What type of tissue culture plates were used (company and model number)? What do the authors mean by “pure free” biofilms?

Lines 116 and 119: Were “media only” controls or other negative controls used in this part of the study? Were CFU/mL counts taken at the beginning of the experiments when tissue culture plates or teeth were inoculated with E. faecalis?

Line 134: Is 5.25% NaOCl a recommended concentration for use in patients?

Line 147: What buffer were cells diluted in for serial dilutions?

Lines 158-159: more details are needed for the SEM (voltages, magnification). How were samples stored prior to imaging?

Line 166: E. faecalis isolates can vary in their biofilm production. How was this strain of E. faecalis chosen? Are there any references the authors can provide for this strain?

Lines 168-173: These experiments are not described in the materials and methods section.

Figure 1: Are these results from three biological replicates?

Line 181: It would be helpful to show this data if the authors have evidence of mature biofilms detaching from surfaces. E. faecalis will form mature biofilms by ~24 hr after inoculation, but biofilms formed in BHI are typically not as large as in other media.

Lines 182-190: It seems unlikely that biofilm cells could be completely disrupted by extensive pipetting. Do the authors know if they broke biofilm clumps completely apart and if not, how this affected CFU/mL plating counts?

Lines 208-218: There are alternative methods for preserving biofilm structures that have been developed. These include a dehydration gradient in ethanol and addition of cationic dyes to help stabilize the matrix. Using these methods will allow the authors to get higher quality images of E. faecalis biofilms using SEM (although I am not suggesting the authors repeat the SEM in this study).

Figure 3: showing the individual data points in addition to the average & error bars would be helpful in evaluating the data.

Line 248: an average of 1.39E1 what? CFU/mL?

Author Response

Reviewer 2

  1. Overall, the experiments seem well designed, although there is not enough information provided in the materials and methods section (see comments below). Additionally, several previous studies have found differing results regarding the killing of E. faecalis by NaOCl. The discussion and conclusions would be significantly strengthened if the authors compared and constrasted their results with these previous studies. Why do the authors think they saw such good inhibition of E. faecalis in this study?

Thank you for the critiques, we hope we can improve the text with your suggestions. To explain the efficacy of HEDP in NaOCl, we are including in the discussion the idea that this is because the combination of a chelating agent with sodium hypochlorite increases the bacterial disinfection efficacy. The ideal chelator should not interfere with sodium hypochlorite. The chelating agent removes the smear layer so that the sodium hypochlorite can access all surfaces. HEDP is the chelator that we have seen does not interact with sodium hypochlorite, and thus, the most efficacious one.

  1. Was the strain of faecalis or the way the assays were set up different than other studies that saw higher rates of resistance?

The strain of Enterococcus faecalis used in our experiments has not being used before in this kind of experiments. We are not sure if our strain may have additional characteristics as compared to previously used strains. The protocol to prepare the mechanically instrumented teeth include both uni, and multiradicular roots. Finally, the protocol to generate best biofilms previous to biofilm decay was two weeks and the media of choice (BHI) may differ from some studies, determining the difficulty to remove the biofilm in our model. In addition to NaOCl combined with EDTA, we included a group to test the combination of sodium hypochlorite with another chelator, HEDP, which has better characteristics that allow it to be used in combination with sodium hypochlorite.

  1. Minor comment: Multiple examples of spaces missing between words: line 26 (roots amputated), line 14 (address requests for reprints to), line 40 (bacteria in), line 51 (Enterococcus faecalis is a), etc.

Thank you for noticing, the text has been reviewed again and changed accordingly.

  1. Line 22: please define PUI acronym here

Thanks, we indicate PUI acronym meaning in line 19.

  1. Line 36: please define NaOCl abbreviation the first time it is used

Thanks, we indicate NaOCl acronym meaning in line 23.

  1. Line 49: Many biofilm bacteria don’t “prefer” necrotic tissue

Thank you for noticing, the text has been modified to “with a preference for necrotic tissues” in line 37.

  1. Line 95: Were the teeth removed from patients and prepared for autoclaving immediately? Or were they stored before autoclaving and biofilm assays? Can the authors provide more information on the patients from which teeth were obtained?

The teeth were collected over a period of time of about 6 months from different dental clinics. Individual teeth were chosen following inclusion criteria described in the manuscript. Once all the samples that satisfied the inclusion criteria were collected, the teeth were mechanically instrumented, followed by autoclaving. Overall the teeth were extracted for different reasons such as orthodontic treatment or decay. 

  1. Line 113: Was E. faecalis CECT 481 originally stored as a freezer stock? How long werw the bacteria kept on agar plates in total?

Yes, the strain used in these assays was stored in the freezes of our collection at -20ºC for several years. To work with a stable culture, after thawing, individual colonies in the recovered bacteria were isolated and reculture for several passages, at least 4, before the initiation of the experiments. Bacteria was kept alive in fresh cultures of Slanetz-Barley for two /three moths before starting with fresh bacteria from the stock again.

  1. Line 116: What type of tissue culture plates were used (company and model number)? What do the authors mean by “pure free” biofilms?

We used Slanetz-Barley agar plates from OXOID catalogue number: PO5018A. The reference has been included in the manuscript text line 108.

  1. Lines 116 and 119: Were “media only” controls or other negative controls used in this part of the study? Were CFU/mL counts taken at the beginning of the experiments when tissue culture plates or teeth were inoculated with E. faecalis?

The Control group in this case is referring to the teeth that were not treated with any irrigant or file activation. In this case, this is the reference to determine the effect of any of the following treatments. The initial inoculum used in the assays was consistently passes the day before to have fresh bacteria and standardized by densitometry using a 0.5 Mc Farland. After inoculation and incubation, the bacteria were recovered as indicated and serialy diluted before plating in Slanetz-Barley plates. Plates were incubated for 2 days at 37ºC before counting the number of colonies and determining the CFU/ml.  Thus, the bacterial count of each sample took place after the treatment was performed.

  1. Line 134: Is 5.25% NaOCl a recommended concentration for use in patients?

Yes, the 5,25% concentration of sodium hypochlorite is recommended to obtain the highest degree of disinfection in a tooth during root canal treatment, although there is no established agreement on what concentration of sodium hypochlorite to use.

  1. Line 147: What buffer were cells diluted in for serial dilutions?

Dentin was extracted from each tooth sample and diluted in Eppendorf tubes with 1mL of PBS. The dilution is described in lines 120 and 136.

  1. Lines 158-159: more details are needed for the SEM (voltages, magnification). How were samples stored prior to imaging?

Thank you for noticing., SEM voltage and magnification re indicated inside the image. We have included this information at the image legend. Voltage was set at 10 kV. Magnification was 2000x. The preparation of the samples is described in materials and methods: lines 169-171.

  1. Line 166: E. faecalis isolates can vary in their biofilm production. How was this strain of E. faecalis chosen? Are there any references the authors can provide for this strain?

The strain was arbitrary used based on our availability. The identity of the strain we selected was the one we had in our stock records. The identity was verified by using a Vitec 2 automated identification. This strain that we selected for our study and we always used the same one. We don´t have any previous publications using this strain, this is our first one.

  1. Lines 168-173: These experiments are not described in the materials and methods section.

This initial experiment was performed to evaluate the behavior of the irrigants against the bacteria from a direct contact with the bacteria and not inside the tooth. It was previously performed before our study.

We include a brief description of these methods in materials and methods lines 109 to 119. 

  1. Figure 1: Are these results from three biological replicates?

These results were obtained from the initial experiment we performed on the irrigants against the isolated bacteria. These results were obtained from an experiment performed in triplicate. No differences were observed.

  1. Line 181: It would be helpful to show this data if the authors have evidence of mature biofilms detaching from surfaces. E. faecalis will form mature biofilms by ~24 hr after inoculation, but biofilms formed in BHI are typically not as large as in other media.

Thank you for your point. In fact, we were observing detaching biofilms after 2 weeks and we were having troubles to standardized conditions in such experiments if we were waiting more time after these two weeks. We agree that this information is interesting, however, to be able to consistently compare conditions, we choose the indicated ones. Future experiments may try to describe the implications of older biofilms in the stablished bacteria colonization and removal from the teeth.

  1. Lines 182-190: It seems unlikely that biofilm cells could be completely disrupted by extensive pipetting. Do the authors know if they broke biofilm clumps completely apart and if not, how this affected CFU/mL plating counts?

We are not completely sure of whether we completely disrupted biofilm by pipetting, however, we were consistently getting the expected results from serial dilutions and from different experiments, which makes our technique reproducible and give consistent information about the effectiveness of bacterial debridement. In any case, the efficacy of HEDP in NaOCl in our experimental model was complete and  not bacteria could be recover from this condition at all.

  1. Lines 208-218: There are alternative methods for preserving biofilm structures that have been developed. These include a dehydration gradient in ethanol and addition of cationic dyes to help stabilize the matrix. Using these methods will allow the authors to get higher quality images of E. faecalis biofilms using SEM (although I am not suggesting the authors repeat the SEM in this study).

Thank you very much for your suggestion. We are in the process of getting access to a new TEM microscope and we hope that we can include your suggestion in future experiments. We the best we could for the given conditions.

  1. Figure 3: showing the individual data points in addition to the average & error bars would be helpful in evaluating the data.

Thank you for the suggestion. We have introduced the suggesting change in order to show individual numbers.

  1. Line 248: an average of 1.39E1 what? CFU/mL?

Thank you for your suggestion, we have introduced CFU/ml in the text.

Reviewer 3 Report

The experiment is quite interesting and wel conducted but not well presented, as well as a concept is missing: 

Indeed e. faecalis and biofilm forms in the canal due to the narrow space which sometimes cannot allow to the irrangants to reach the biofilm and to allow the microbicidal effect. 

In addition, the edta acts as adjuvant in removal biofilm, which protects the microbial community. 

Methods, are very poorly described: ok about the morphological protocol, but what about the quantification protocol?? please rewrite clearly choerently. 

Author Response

Reviewer 3’s comments:

  1. The experiment is quite interesting and well conducted but not well presented, as well as a concept is missing: Indeed e. faecalis and biofilm forms in the canal due to the narrow space which sometimes cannot allow to the irrigants to reach the biofilm and to allow the microbicidal effect. In addition, the edta acts as adjuvant in removal biofilm, which protects the microbial community.

Thank you to the reviewer for appreciating the value of some challenging experiments. We have tried to improve the presentation of the data (see figure 3). We are open to try to improve it following your suggestions.

  1. Methods are very poorly described: ok about the morphological protocol, but what about the quantification protocol?? please rewrite clearly coherently. 

Thank you for your suggestion. In order to clarify reviewer´s concerns, bacteria treatment section has been thoroughly revised (lines 109 to 119). We have tried to also improve the description of the electron microscopy in materials and methods in lines 165-171.

Round 2

Reviewer 1 Report

Certainly, there are significant improvement for the revised manuscript structure and contents. However, authors still need clarification and should carefully read again using fresh eyes for the typos, redundancy, abbreviation rules, and grammar.

Figure1. Still missing vertical axis-species information, there are no difference between A) & B) regarding the results. B) represented a biofilm, however the CFU is lesser than the A), which need some explanation. Based on author's object, NaOCl has shown limitations as an endodontic irrigant, however the both graphs telling us that NaOCl is simply working great. It is not compelling data for author's perspective.

Figure 2. poor image quality could be improved

Author Response

Reviewer 1’s comments:

  1. Certainly, there are significant improvement for the revised manuscript structure and contents. However, authors still need clarification and should carefully read again using fresh eyes for the typos, redundancy, abbreviation rules, and grammar.

Thank you for noticing. We have tried our best to fulfill all concerns. We have read the whole manuscript to find typos, redundancy, abbreviation rules, and grammar. We have highlighted the changes along the text in yellow.

  1. Still missing vertical axis-species information, there are no difference between A) & B) regarding the results. B) represented a biofilm, however the CFU is lesser than the A), which need some explanation. Based on author's object, NaOCl has shown limitations as an endodontic irrigant, however the both graphs telling us that NaOCl is simply working great. It is not compelling data for author's perspective.

We are sorry if we were not clear enough before. As explained in the Figure legend and in the text, A) represents the direct antibacterial effect of the different irrigants on fresh bacteria (no biofilm or teeth are present here at this point). In B) however, we have generated E. faecalis mature biofilm in vitro (no teeth), to try to determine the direct effect of the irrigant on mature biofilm outside of the teeth. We have introduced “bacteria forming mature biofilm” in the figure legend to try to be more specific.

The conditions for A) and B) are different (see materials and methods lines 105 to 129). We agree with reviewer that the data are different in this figure as compare to figure 3. However, in Figure 1, the experiment is performed in the absence of teeth, as compared to Figure 3. In our opinion, comparing the 3 assays (bacteria alone, bacteria forming biofilm and biofilm inside root canals) allow us to visualize that the complexity of removing biofilm is due to the presence on biofilm at the root canal and not the bacteria or the biofilm alone.

Figure 2. poor image quality could be improved

Thank you for the suggestion. As we explain in the text, lines 225-230, this image has challenging issues and was the best possible image using the indicated instrument that we had available. We believe that the image contains enough information to be considered in the manuscript.

We already addressed this issue in our previous answer to reviewer 2:

Lines 208-218: There are alternative methods for preserving biofilm structures that have been developed. These include a dehydration gradient in ethanol and addition of cationic dyes to help stabilize the matrix. Using these methods will allow the authors to get higher quality images of E. faecalis biofilms using SEM (although I am not suggesting the authors repeat the SEM in this study).

Thank you very much for your suggestion. We are in the process of getting access to a new TEM microscope and we hope that we can include your suggestion in future experiments. We the best we could for the given conditions.

Reviewer 3 Report

Authors complied to the suggestions. 

Manuscript can be now accepted 

Author Response

We thank you to the reviewer for the comments and suggestions to improve the manuscript.